# Strobilurin biosynthesis in Basidiomycete fungi

Risa Nofiani[1,2], Kate de Mattos-Shipley[1], Karen E. Lebe[3,4], Li-Chen Han [1], Zafar Iqbal[1,5], Andrew M. Bailey[6], Christine L. Willis [1], Thomas J. Simpson[1] & Russell J. Cox [1,3,4]

Strobilurins from fungi are the inspiration for the creation of the β-methoxyacrylate class of agricultural fungicides. However, molecular details of the biosynthesis of strobilurins have remained cryptic. Here we report the sequence of genomes of two fungi that produce strobilurins and show that each contains a biosynthetic gene cluster, which encodes a highly reducing polyketide synthase with very unusual C-terminal hydrolase and methyltransferase domains. Expression of *stpks1* in *Aspergillus oryzae* leads to the production of prestrobilurin A when the fermentation is supplemented with a benzoyl coenzyme A (CoA) analogue. This enables the discovery of a previously unobserved route to benzoyl CoA. Reconstruction of the gene cluster in *A. oryzae* leads to the formation of prestrobilurin A, and addition of the gene *str9* encoding an FAD-dependent oxygenase leads to the key oxidative rearrangement responsible for the creation of the β-methoxyacrylate toxophore. Finally, two methyl-transferases are required to complete the synthesis.

[1] School of Chemistry, University of Bristol, Cantock's Close, Bristol BS8 1TS, UK. [2] Department of Chemistry, Faculty of Mathematics and Natural Sciences, University of Tanjungpura, Jl. Prof. Dr. H. Hadari Nawawi, Pontianak 78124, Indonesia. [3] Institute for Organic Chemistry, Leibniz Universität Hannover, Schneiderberg 1B, 30167 Hannover, Germany. [4] BMWZ, Leibniz Universität Hannover, Schneiderberg 38, 30167 Hannover, Germany. [5] Department of Agricultural Chemistry, Faculty of Nutrition Sciences, The University of Agriculture, Peshawar 25130 Khyber Pakthunkhwa, Pakistan. [6] School of Biological Sciences, University of Bristol, 24 Tyndall Avenue, Bristol BS8 1TH, UK. Correspondence and requests for materials should be addressed to R.J.C. (email: russell.cox@oci.uni-hannover.de)

Strobilurin A **1**[1] and oudemansin A **2**[2] (also known as mucidin) are antifungal polyketides produced by various Basidiomycete fungi (Fig. 1a). The key β-methoxyacrylate toxophore targets the Qo site of complex III of the mitochondrial electron transport chain and prevents adenosine triphosphate synthesis[3]. The major class of β-methoxyacrylate agricultural fungicides were developed from the structures of **1** and **2** with the aim of increasing photo-stability and selectivity. Thus, compounds such as azoxystrobin **3** (Syngenta) and Kresoxim methyl **4** (BASF) are among the most widely used fungicides worldwide. This class of antifungals are used as effective treatments against a broad range of destructive fungal plant pathogens and make significant contributions to food security[4,5]. The strobilurin fungicides are estimated to have been worth $3.4 billion in 2015 and they make up 25% of the fungicide market and 6.7% of the total crop protection market. Numerous functionalised strobilurin natural products are known[6] in which the aromatic ring is hydroxylated (e.g. strobilurin F **5**)[7] and chlorinated (e.g. strobilurin B **6**)[1] and the hydroxyl groups can be, in turn, methylated or prenylated (e.g. strobilurin G **7**)[7,8]. Other compounds such as the formally reduced congener bolineol **8**[9] have also been reported. Surprisingly, despite the high level of interest in the strobilurins as agricultural fungicides, remarkably little is known of their detailed biosynthesis.

Isotopic feeding experiments have shown that strobilurin A **1**[10] and oudemansin A **2**[11] are polyketides, unusually derived from a benzoate starter unit derived by the degradation of phenylalanine via cinnamate. In fungi, polyketides are produced by iterative polyketide synthase (PKS) enzymes that extend acyl coenzyme A (CoA) starter units using decarboxylative Claisen-type reactions with malonyl CoA[12]. The isotopic labelling patterns of **1** and **2** are consistent with three extensions of a benzoate starter unit and C-methylation of the chain (Fig. 1b) using S-adenosyl methionine (SAM). An unusual oxidative rearrangement is involved in the formation of the core β-methoxyacrylate, and O-methylations must also occur to give strobilurin A **1** and oudemansin A **2**. However, the molecular basis for these reactions has remained unexplored.

Here we reveal the biosynthetic gene cluster (BGC) responsible for the construction of the strobilurins and reconstruct the biosynthesis of **1** in the heterologous host Aspergillus oryzae. This reveals an unusual type of fungal PKS and the enzyme responsible for the key oxidative rearrangement reaction.

## Results

**The strobilurin BGC.** Strobilurins have been reported mainly as products of Basidiomycete fungi such as Strobilurus tenacellus, but one apparent Ascomycete, Bolinea lutea, is also reported as a producer of a range of strobilurins and related compounds such as bolineol **8**[8]. In our hands S. tenacellus produced strobilurin A **1** (typically 30 mg L$^{-1}$ under optimised conditions), B **6** and G **7**, and B. lutea F23523[7] reliably produced strobilurin B **6** (ca 1 mg L$^{-1}$) in shake-flask culture. We sequenced the genomes of each organism to produce draft genomes using standard Illumina paired-end methodology (Supplementary Figure 1, Supplementary Tables 1–3). The S. tenacellus assembly produced a draft genome of 40.5 Mb with an $N_{50}$ of 121.3 kb, while the B. lutea genome had a total size of 42.6 Mb and an $N_{50}$ of 63.5 kb. The data are consistent with a conclusion that the organism previously known as B. lutea is, in fact, a Basidiomycete, and sequence analysis of its internal transcribed spacer sequence showed it to be most likely a previously unreported member of the Strobilurus family (Supplementary Figure 8). We thus rename it Strobilurus lutea.

Since fungal BGCs are usually relatively compact (often <50 kb) the assemblies were considered sufficient to search for genes potentially involved in the biosynthesis of **1** and its derivatives. Anti-SMASH software[13] identified two potential PKS-encoding BGCs in each genome (Supplementary Figure 6). The first of these encodes the expected type I iterative PKS typical of fungi[12], and nearby are genes predicted to encode proteins involved in phenylalanine metabolism (for example, a phenylalanine ammonia lyase, PAL), methylation and redox reactions. The second cluster in each fungus lacks these types of genes and was not considered further. Our initial analysis showed that the S. lutea BGC is contained on one scaffold (SL-298) of 106 kb. In contrast, the homologous PKS cluster in S. tenacellus is located on a smaller scaffold of only 30 kb (ST-273), suggesting that although this scaffold aligns well with the S. lutea cluster it may only represent a partial cluster. Data from the longer SL-298 scaffold was then used to search the S. tenacellus data for the missing genes and scaffold-195 (ST-195, ca 145 kb) was identified as containing a number of homologues. Oligonucleotide primers designed based on the terminal sequences of ST-195 and ST-273 were used in a PCR reaction with S. tenacellus genomic DNA as the template and this yielded a ca 4 kb product, which was sequenced. The sequence closed the gap between ST-273 and ST-195 demonstrating that these two scaffolds are genuinely adjacent genomic loci and represent separate parts of the same gene cluster. The total size of the combined S. tenecellus scaffold was 179.2 kb (Fig. 2).

**a**

Strobilurin A **1**

Oudemansin A **2**

Azoxystrobin **3**

Kresoxim methyl **4**

**5**

**6**

**7**

Bolineol **8**

**b**

Oudemansin A

**Fig. 1** Structures of key natural (**1**, **2**, **5**–**8**) and synthetic (**3**, **4**) strobilurins. **a** Compounds **1**–**3** and **5**–**7** feature the key β-methoxyacrylate toxophore, while **8** is reduced. Compound **4** possesses a chemically similar methoxyiminoacetate toxophore; **b** incorporation of biosynthetic precursors into oudemansin A. Origin of atoms as indicated by bold bonds, and open and closed circles

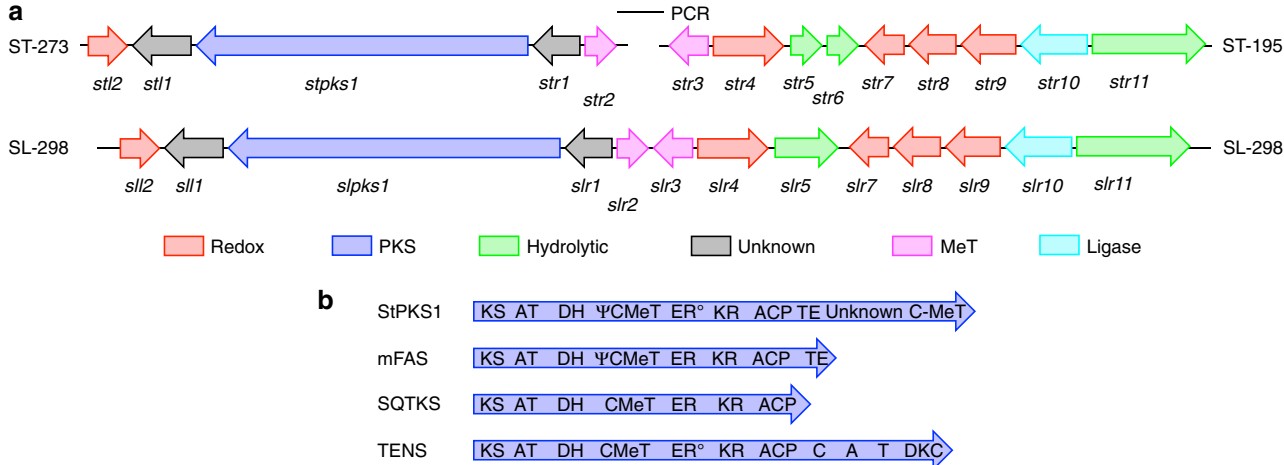

**Fig. 2** Map of the strobilurin biosynthetic gene clusters in *S. tenacellus* and *S. lutea*. **a** Annotated ca 47 kb biosynthetic gene clusters contained in the *S. tenacellus* and *S. lutea* genomes (not to scale); **b** domain structure of known fungal highly reducing polyketide synthases. KS: keto-synthase, AT: acyl-transferase, DH: dehydratase, *C*-MeT: *C*-methyltransferase; ΨC-MeT: non-functional *C*-MeT, ER: enoyl-reductase, ER°: non-functional ER, KR: keto-reductase, ACP: acyl carrier protein, TE: thiolesterase, C: condensation domain, A: adenylation domain, T: thiolation domain, DKC: Dieckmann cyclase, StPKS1: *S. tenacellus* polyketide synthase 1, mFAS: mammalian fatty acid synthase, SQTKS: squalestatin tetraketide synthase, TENS: tenellin synthetase

Basidiomycete genes usually contain numerous intron sequences, making automated gene-calling difficult. However, using predominantly manual methods, a total of 62 potential protein-encoding genes were identified and annotated between the overlapping scaffolds (Supplementary Table 5). In order to further clarify the data, Illumina transcriptome sequencing was performed using mRNA purified from *S. tenacellus* under strobilurin-producing conditions (Supplementary Figure 7). This refined the procedure of annotating the genes by confirming the transcriptional start and stop positions and the precise positions of introns, allowing correct coding sequences to be determined. Relative to the PKS, genes located downstream of the PKS are denoted *l* (left) and genes upstream are denoted *r* (right), and genes of *S. tenacellus* are denoted *st*, while genes of *S. lutea* are denoted *sl*. Thus, the gene *slr11* is the eleventh identified gene to the right of the *S. lutea* PKS, and it encodes a PAL (Table 1).

We focussed our investigations on likely secondary metabolism genes immediately upstream and downstream of the PKS gene that appeared to be potentially involved in secondary metabolism (Fig. 2a). We therefore set the approximate cluster boundaries as *stl2* downstream of the PKS and *str11* upstream of the PKS, defining a ca 47 kb sequence for more detailed analysis. Resistance to strobilurins is known to be mediated by mutations in mitochondrial cytochrome b (encoded by *CYTB*)[14,15]. Whilst no homologues of *CYTB* were found within the cluster, or on the larger ST-195/273 or SL-298 scaffolds, we note that the mitochondrial *CYTB* shows motifs characteristic of being strobilurin resistant (see Supplementary Methods).

The PKS itself is encoded by a single large multi-exon coding sequence. Initial examination showed this to contain keto-synthase, acyl-transferase (AT), dehydratase (DH), *C*-methyltransferase (*C*-MeT), enoyl-reductase (ER), keto-reductase (KR) and acyl carrier protein (ACP) domains, showing that it belongs to the fungal highly reducing class of PKS (hr-PKS, Fig. 2b)[12]. Very unusually, the ACP is followed by a partial αβ-hydrolase domain closely related to the first 100 residues of the C-terminal thiolesterase (TE) of mammalian fatty acid synthase (mFAS, 33% identical, 57% similar). This peptide sequence contains the conserved serine nucleophile (S2240) but not the cognate aspartate and histidine required by αβ-hydrolases. This sequence is followed by ca 300 residues with no significant sequence or structural homology to any known protein, although there is

homology to predicted hr-PKS from other Basidiomycetes such as *Stereum hirsutum* (XP_007303267.1) and *Gloeophyllum trabeum* (XP_007863729.1, 37–41% identity, 57–61% similarity). The final 220 residues are homologous to SAM-dependent *C*-methyltransferases, including the CurJ *C*-MeT from curacin biosynthesis (32% identical, 48% similar)[16]. Detailed protein sequence comparisons with mFAS, the squalestatin tetraketide synthase (SQTKS)[17] and tenellin synthetase (TENS)[18] suggested that the *C*-MeT domain located between the DH and KR domains is probably inactive due to mutations in the SAM-binding motif, as is the ER, which is more similar to the inactive ER domain of TENS (Supplementary Figures 10–11). However, sequence analysis suggested that the C-terminal *C*-MeT domain is likely to be active, as the SAM-binding site appears to be intact. The presence of all these domains as a single peptide was confirmed from the transcriptome data that showed a single transcript encoding the entire 306 kDa protein (Supplementary Figure 12).

Other genes were functionally annotated by extensive manual comparisons to known genes from fungi, and while tentative functions could be proposed for most translated proteins, the roles of the *r1* and *l1* gene products remain enigmatic. A hydrolytic function encoded by *slr5* as a single putative gene in the *S. lutea* genome appears to be formed as a fusion of two separate hydrolases encoded by *str5* and *str6* in the *S. tenacellus* genome. However, all other genes are encoded in the same relative positions and orientations in each cluster.

**Heterologous expression of early pathway genes.** We investigated expression of key genes from the cluster in *A. oryzae*[19]. Since the PKS is predicted to contain 28 introns, and the Ascomycete *A. oryzae* is unlikely to process all of these correctly[20], we amplified intron-free fragments of *stpks1* from cDNA, and reconstructed the entire 8.5 kb coding region of the PKS using recombination in yeast. The recombined intron-free *stpks1* was cloned into the fungal expression vector pTYGS-Arg[21], which was, in turn, transformed into *A. oryzae* strain NSAR1, which harbours four auxotrophic lesions on its genome allowing the introduction of up to four DNA fragments in parallel[22–24]. The pTYGS vector contains an *argB* gene, which complements one of the auxotrophies, allowing selection on minimal media with appropriate supplements. The cloned gene of interest is expressed

**Table 1 Detected genes located adjacent to *pks1***

| Gene | *Strobilurus lutea* predicted protein function | AA | Gene | *Strobilurus tenacellus* predicted protein function | AA | % Ident. | Predicted cofactor |
|---|---|---|---|---|---|---|---|
| *sll2* | SDR | 335 | *stl2* | SDR | 340 | 97 | NAD(P) |
| *sll1* | Hypothetical protein | 499 | *stl1* | Hypothetical protein | 498 | 84 | – |
| *slpks1* | PKS | 2815 | *stpks1* | PKS | 2824 | 93 | NAD(P), SAM |
| *slr1* | Hypothetical protein | 404 | *str1* | Hypothetical protein | 207 | 82 | – |
| *slr2* | Methyltranferase | 269 | *str2* | Methyltransferase | 275 | 95 | SAM |
| *slr3* | Methyltranferase | 332 | *str3* | Methyltransferase | 287 | 94 | SAM |
| *slr4* | GMCO | 600 | *str4* | GMCO | 632 | 95 | FAD |
| *slr5* | Hydrolase | 630 | *str5* | Hydrolase 1 | 278 | – | – |
|  |  |  | *str6* | Hydrolase 2 | 304 | – | – |
| *slr7* | Aldoketoreductase | 336 | *str7* | Aldoketoreductase | 336 | 96 | NAD(P) |
| *slr8* | NHI | 416 | *str8* | NHI | 425 | 77 | Fe, α-ketoglutarate |
| *slr9* | FDO | 452 | *str9* | FDO | 395 | 87 | FAD, NAD(P) |
| *slr10* | CoA ligase | 576 | *str10* | CoA Ligase | 637 | 89 | ATP |
| *slr11* | PAL | 991 | *str11* | PAL | 702 | 91 | – |

See Supplementary Table 5 for a full list of genes found on the entire contig. Shaded rows indicate genes shown to be involved in the biosynthesis of strobilurin or bolineol by heterologous expression. Predicted functions from NCBI-conserved domain analysis
SDR: short-chain dehydrogenase, PKS: polyketide synthase, SAM *S*-adenosyl methionine, GMCO: glucose-methanol-choline oxidoreductase, PAL: phenylalanine ammonia lyase, NAD(P): nicotinamide adenine dinucleotide (phosphate), FAD: flavin adenine dinucleotide, ATP: adenosine triphosphate

under the control of the *amyB* promoter ($P_{amyB}$), which is induced, by starch and maltose[25].

In initial experiments, expression of *stpks1* alone in *A. oryzae* led to no significant changes in the metabolite profiles of transformed vs. untransformed strains (Table 2, expt 1). However, since benzoyl CoA is likely to be the starter unit for the biosynthesis of the expected strobilurin polyketide, and it is unknown whether *A. oryzae* can make this intermediate, we then supplemented fermentations with either benzoic acid or the *N*-acetyl cysteamine thiolester of benzoic acid (benzoyl SNAC 9)[26], which is a benzoyl CoA mimic. In these experiments a new compound 11 was produced in the presence of benzoyl SNAC 9 (Fig. 3a), but not benzoic acid 10 (Fig. 3b, Supplementary Figures 31–33) or in its absence (Fig. 3c) or in the control (Fig. 3d).

Analysis by liquid chromatography mass spectrometry (LCMS) showed that the new compound 11 had a mass ($m/z$ 215 [M]H$^+$) consistent with a methylated tetraketide. The compound was isolated as a yellow powder (20 mg L$^{-1}$). High-resolution mass spectrometry (HRMS, 213.0915, [M-H]$^-$) confirmed the molecular formula as $C_{14}H_{13}O_2$ (calc. 213.0916). The structure was solved by analysis of one-dimensional and two-dimensional (2D) nuclear magnetic resonance (NMR) spectroscopy data (Fig. 3e, Supplementary Figure 34) that showed the presence of a monosubstituted benzene, two separate alkene spin systems with three and two protons respectively and a vinylic methyl group. Heteronuclear multiple bond correlation (HMBC) analysis showed the structure to be the triene 11, and observation of nOe between H-3 and H-6 suggested the *Z*-configuration of the

central olefin. The other two olefins were determined to be *E* from large vicinal coupling constants (ca 15.4 Hz).

The structure and triene geometry was confirmed by total synthesis (Fig. 4). Briefly, the phenyl pyrone 12[27] was reduced to the unsaturated lactone 13 that was then ring-opened by elimination to give the *E,Z* triketide 14. Activation to the mixed anhydride/carbonate and reduction at low temperature gave the expected primary alcohol 15, which was oxidised to the corresponding aldehyde and subjected to Horner Wadsworth Emmons homologation. The resulting *E,Z,E* triene methyl ester was hydrolysed to give the corresponding tetraketide acid 11, which was identical (LCMS and NMR analysis, supplementary figure 34) to the material isolated from *A. oryzae*. Since this compound appears to be the first isolable intermediate in strobilurin biosynthesis we name it prestrobilurin A.

The apparent inability of *A. oryzae* to produce benzoyl CoA 16, or convert benzoic acid 10 to benzoyl CoA 16 then allowed us to probe the biosynthesis of this starter unit by using the PKS as a reporter for its presence. A second series of vectors was therefore constructed exploiting the adenine auxotrophy of *A. oryzae* NSAR1 and which contained combinations of *str10*, *str8* and *str11*, which encode a CoA ligase, a non-haem iron oxygenase and a PAL respectively (see Supplementary Methods). Co-expression of *stpks1* with *str10* in *A. oryzae* (Table 2, expt 4) once again did not produce any new compounds, but feeding benzoic acid 10 to the fermentation resulted in the restoration of prestrobilurin A 11 production (Table 2, expt 5). Next, *stpks1* was co-expressed with *str10* and *str8* and again, in the absence of feeding this did not produce 11 (Table 2, expt 6). However, supplementation of either

**Table 2 Summary of heterologous expression experiments in *A. oryzae* designed to probe the biosynthesis of 1**

| Construct | *stpks1* PKS | *str11* PAL | *str8* NHI | *str10* CoA ligase | *str9* FDO | *str4* GMCO | *stl2* SDR | *str2* Met1 | *str3* Met2 | 9 | 10 | 17 | Products |
|---|---|---|---|---|---|---|---|---|---|---|---|---|---|
| Expt 1 | ✓ | – | – | – | – | – | – | – | – | – | – | – | Nothing |
| Expt 2 | ✓ | – | – | – | – | – | – | – | – | – | ✓ | – | Nothing |
| Expt 3 | ✓ | – | – | – | – | – | – | – | – | ✓ | – | – | **11** |
| Expt 4 | ✓ | – | – | ✓ | – | – | – | – | – | – | – | – | Nothing |
| Expt 5 | ✓ | – | – | ✓ | – | – | – | – | – | – | ✓ | – | **11** |
| Expt 6 | ✓ | – | ✓ | ✓ | – | – | – | – | – | – | – | – | Nothing |
| Expt 7 | ✓ | – | – | – | – | – | – | – | – | – | – | ✓ | Nothing |
| Expt 8 | ✓ | – | ✓ | ✓ | – | – | – | – | – | – | – | ✓ | **11** |
| Expt 9 | ✓ | – | ✓ | ✓ | – | – | – | – | – | – | ✓ | – | **11** |
| Expt 10 | ✓ | ✓ | ✓ | ✓ | – | – | – | – | – | – | – | – | **11** |
| Expt 11 | ✓ | ✓ | ✓ | ✓ | ✓ | ✓ | ✓ | ✓ | ✓ | – | – | – | **1, 8, 11, 21** |
| Expt 12 | ✓ | ✓ | ✓ | ✓ | ✓ | ✓ | – | ✓ | ✓ | – | – | – | **1, 21** |
| Expt 13 | ✓ | ✓ | ✓ | ✓ | – | ✓ | – | ✓ | ✓ | – | – | – | **11** |
| Expt 14 | ✓ | – | – | – | – | ✓ | – | ✓ | ✓ | ✓ | – | – | **11** |
| Expt 15 | ✓ | ✓ | ✓ | ✓ | ✓ | – | ✓ | – | – | – | – | – | **21, 22** |
| Expt 16 | ✓ | ✓ | ✓ | ✓ | ✓ | – | – | – | ✓ | – | – | – | **21, 22** |
| Expt 17 | ✓ | ✓ | ✓ | ✓ | ✓ | – | ✓ | ✓ | ✓ | – | – | – | **1, 8, 21, 22** |
| Expt 18 | ✓ | ✓ | ✓ | ✓ | ✓ | – | ✓ | ✓ | ✓ | – | – | – | **8** |

The indicated genes were co-expressed in *A. oryzae* in the presence of the indicated compounds
NHI: non-haem iron oxygenase, PAL: phenylalanine ammonia lyase, PKS: polyketide synthase, MeT: methyltransferase, GMCO: glucose-methanol-choline oxidase, FDO: flavin-dependent oxygenase, SDR: short-chain dehydrogenase/reductase

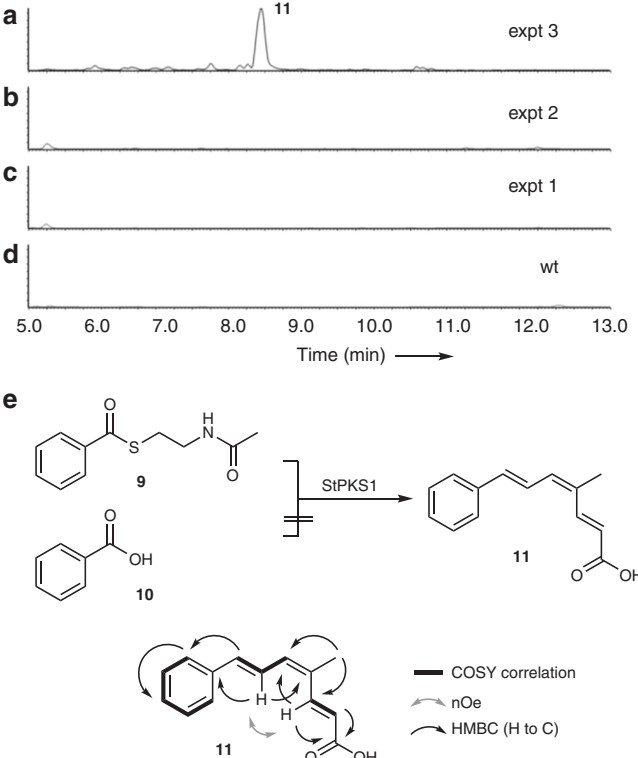

**Fig. 3** In vivo activity of the strobilurin PKS encoded by *stpks1*. LCMS chromatograms (Diode Array Detector, DAD, 200–600 nm, arbitrary units) of extracts of *A. oryzae* NSAR1: **a** *stpks1* + benzoyl SNAC **9**; **b** *stpks1* + benzoic acid **10**; **c** *stpks1* alone; **d** WT NSAR1, LCMS method 1, LCMS liquid chromatography mass spectrometry. See Table 2 for details; **e** production of prestrobilurin A **11** by StPKS1. StPKS1 accepts benzoyl SNAC but not benzoic acid. Summary of 2D nuclear magnetic resonance (NMR) data

**Fig. 4** Total synthesis of prestrobilurin A 11. Reagents and conditions: i PhCHO, $^{n}$BuLi, $^{i}$Pr$_2$NH, HMPA, THF, −78 °C; ii KOH (aq); iii Tf$_2$O, $^{i}$Pr$_2$EtN, CH$_2$Cl$_2$, −78 °C; iv Pd(PPh$_3$)$_4$, Et$_3$SiH, DMF, 60 °C; v Bu$_4$NF, THF; vi EtOCOCl, Et$_3$N, THF, 0 °C; vii NaBH$_4$, MeOH, −78 °C; viii Dess-Martin periodinane, CH$_2$Cl$_2$, RT; ix (MeO)$_2$PCH$_2$CO$_2$Me, NaH, THF, 0 °C to RT; x NaOH, THF, H$_2$O. RT room temperature

cinnamic acid **17** (Table 2, expt 8) or benzoic acid **10** (Table 2, expt 9), restored production of **11** (see supplementary methods). Finally, the *stpks1*, *str10*, *str8* and *str11* genes were co-expressed (Table 2, expt 10). In this case prestrobilurin A **11** was produced without the need to add intermediates, presumably because the PAL can convert endogenous phenylalanine **18** to cinnamate **17**

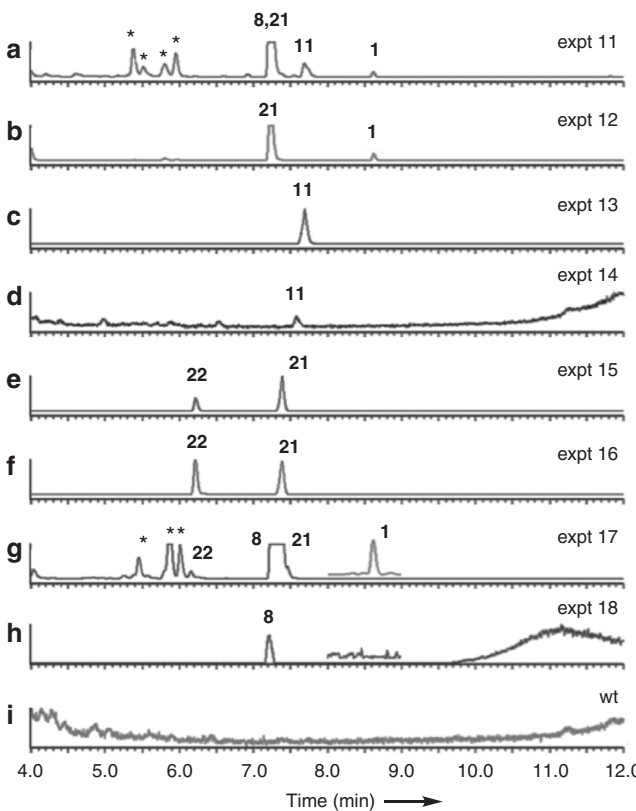

**Fig. 5** Deduced biosynthetic pathway to benzoyl CoA in *S. tenacellus*. The pathway converts phenylalanine **18** to benzoyl CoA **16**, which is then the substrate for StPKS1 to give prestrobilurin A **11**

(Fig. 5). Thus, the plasmid pTYGS-Ade *str8* + *str10* + *str11* acts as an effective source of in vivo benzoyl CoA.

**Heterologous expression of later pathway genes**. In order to investigate the later steps of strobilurin biosynthesis, additional genes were cloned into *A. oryzae* expression vectors (see supplementary methods). We reasoned that *str2* and *str3*, which encode SAM-dependent methyltransferases, are probably responsible for the addition of the two *O*-methyl groups of **1**. Oxidative rearrangement of **11** is required to create the skeleton of **1** and this may be achieved by either the glucose-methane-choline oxidase (GMCO) encoded by *str4* or the flavin-dependent oxidase (FDO) encoded by *str9*. Finally, production of bolineol **8**, observed in *S. lutea*, requires a reduction that may be achieved by the short-chain dehydrogenase (SDR) encoded by *stl2*.

In an initial heterologous experiment the full complement of genes, including the benzoyl CoA-production system (*str8*, *str10* and *str11*), were transferred to *A. oryzae* (Table 2, expt 11). This resulted in production of **11** and **1** (retention time [Rt] 8.4 min, 2.6 mg L⁻¹, Fig. 6a) that was isolated and identified by full NMR analysis (Supplementary Figures 35–39). A new compound (Rt = 7.1 min, $C_{13}H_{15}O_2$ [M + H]⁺ calc. 203.1072, measured 203.1071, Fig. 6a) was also isolated (90 mg L⁻¹) and purified and shown to be the 5*E*,3*Z* carboxymethyldiene **21**, which has been previously synthesised during the total synthesis of strobilurin A **1** (Fig. 7)[28]. Bolineol **8** was detected to coelute with **21** by its distinctive [M + H]⁺ (247.1) and [M + Na]⁺ (269.2) ions.

In a second experiment (Table 2, expt 12, Fig. 6b) the SDR was omitted and this led to a very similar result, but lacking bolineol **8**. Omission of the FDO encoded by *str9* (Table 2, expt 13, Fig. 6c) led to the production of only prestrobilurin A **11**. The same result was achieved if the benzoate genes were omitted and the fermentation was supplemented with benzoyl SNAC **9** (Table 2, expt 14, Fig. 6d), although **11** is produced in much lower titre in this case.

Next, the methyltransferases and GMCO were omitted (Table 2, expt 15, Fig. 6e). This led to the formation of **21**, and also a new compound **22** (Rt 5.9 min, 30 mg L⁻¹), which was shown to be $C_{14}H_{16}O_3$ ([M-H]⁻ calc. 231.1021, measured 231.1021) by HRMS (Fig. 7). Isolation and full structure determination by NMR (Supplementary Figures 42–43) confirmed the new compound **22** to be desmethylbolineol. Desmethylbolineol **22**

**Fig. 6** LCMS evaporative light-scattering (ELS) chromatograms of organic extracts of *A. oryzae* NSAR1 expression strains (arbitrary units). **a** *A. oryzae* NSAR1 + *stpks1* + *str2* + *str3* + *str4* + *str11* + *str8* + *str10* + *str9* + *stl2*; **b** *A. oryzae* NSAR1 + *stpks1* + *str2* + *str3* + *str4* + *str11* + *str8* + *str10* + *str9*; **c** *A. oryzae* NSAR1 + *stpks1* + *str2* + *str3* + *str4* + *str11* + *str8* + *str10*; **d** *A. oryzae* NSAR1 + *stpks1* + *str2* + *str3* + *str4* + benzoyl SNAC **9**; **e** *A. oryzae* NSAR1 + *stpks1* + *str11* + *str8* + *str10* + *str9* + *stl2*; **f** *A. oryzae* NSAR1 + *stpks1* + *str11* + *str8* + *str10* + *str9*; **g** *A. oryzae* NSAR1 + *stpks1* + *str2* + *str3* + *str11* + *str8* + *str10* + *str9*, inset trace shows extracted ion chromatogram for *m/z* 259.1; **h** *A. oryzae* NSAR1 + *stpks1* + *str2* + *str3* + *str11* + *str8* + *str10* + *str9* + *stl2*, inset trace shows extracted ion chromatogram for *m/z* 259.1; **i** untransformed *A. oryzae* NSAR1. *Unrelated compounds. LCMS method 2. See Supplementary Figure 55 for additional data for expt 17

was smoothly converted to bolineol **8** by treatment with trimethylsilyldiazomethane[29] and the spectroscopic data were shown to be identical. Further omission of the SDR (Table 2, expt 16, Fig. 6f) gave the same result, showing that formation of **22** does not require the SDR and must represent a shunt pathway in *A. oryzae* in the absence of the late-acting methyltransferases. Omission of the GMCO and SDR (Table 2, expt 17, Fig. 6g) resulted in extremely unhealthy *A. oryzae* cultures, which were difficult to select and grow (Supplementary Figure 55). Extraction of these cultures revealed the formation of bolineol **8** as the major compound, together with lesser amounts of **21** and **22**. Strobilurin A **1** was detected by its distinctive mass spectrum, but in very low titre. Reinclusion of the SDR (Table 2, expt 18, Fig. 6h) produced **8** only, with no production of **1**. No compounds were produced in the control experiment (Fig. 6i).

Finally, *str9* was codon-optimised and expressed in *Escherichia coli* in N-terminal his-tagged form and purified in soluble form. The *holo*-protein, containing the FAD cofactor (verified by ultraviolet spectroscopy), was confirmed by mass spectrometry. Initial in vitro assays using purified prestrobilurin **11**, purified Str9 and NAD(P)H in various buffers in air were performed, but

**Fig. 7** Oxidative rearrangement of **11** on the pathway to **1** and shunts leading to observed compounds **21** and **22**. **a** Deduced pathway from prestrobilurin **11** to strobilurin A **1**. **b** Similar oxidative rearrangements observed during the biosynthesis of aspyrone **23** and bartanol **24**

no oxidised and rearranged product **26** could be observed by LCMS analysis. We reasoned that this was most likely due to the observed insolubility of **11** in the aqueous buffer system. However, use of a system containing a high ethanol concentration, was more successful and a product corresponding to **26** was clearly observed by LCMS (Supplementary Figure 53).

## Discussion

Our results show that biosynthesis begins with construction of benzoyl CoA **16** by step-wise elimination of ammonia from phenylalanine **18** (*str11*), oxygenation (*str8*) and retro-Claisen reaction to form benzoic acid **10**, which is activated to its CoA thiolester **16** by a dedicated CoA ligase (*str10*, Fig. 5). Benzoyl CoA **16** then forms the starter unit for polyketide biosynthesis. Recent work by our group, and the group of Tang has shown that benzoyl CoA also forms the starter unit for the biosynthesis of the squalestatin hexaketide in Ascomycetes[30,31]. Although the two fungal pathways for the formation of benzoyl CoA share a common PAL enzyme, the squalestatin pathway involves formation of cinnamoyl CoA early in the pathway, while the strobilurin pathway is not consistent with the formation of cinnamoyl CoA as an intermediate, but appears to involve direct oxidative conversion of cinnamic acid **17** to benzoic acid **10**.

Use of benzoate (and other non-acetate) starter units by fungal PKS is rare. Starter unit selection is presumably controlled by the AT domain of the PKS, however in the absence of structural data it is not yet possible to determine the basis of this selectivity. The strobilurin PKS appears to contain an inactive enoyl-reductase domain (ER°) similar to the lovastatin nonaketide synthase[32] and the TENS[33], but no gene encoding a *trans*-acting enoyl-reductase is

present (e.g. *lovC* or *tenC*) and this is consistent with its synthesis of a triene. On the other hand, the strobilurin PKS is highly unusual among fungal hr-PKS in containing catalytic domains located after the ACP domain[12]. The presence of non-canonical catalytic domains such as *C*-MeT and reductive release domains is common for non-reducing PKS (e.g. methylorcinaldehyde synthase)[34], but is more unusual for fungal hr-PKS, although non-ribosomal peptide[35] and carnitine acyl-transferase[36] mechanisms are known. The observed hydrolase domain of the strobilurin PKS may be responsible for release of prestrobilurin A **11**, while the unique C-terminal methyltransferase probably attaches the C-4 methyl group as sequence analysis suggests the more normally positioned *C*-MeT is inactive. Further work will be required to verify this hypothesis.

The strobilurins contain a highly unusual *E,Z,E* triene and the origin of this motif has been hitherto unexplained. Polyketides are already known, which have *Z*-olefins, for example, borrelidin, FR901464 and fostriecin. In the case of the modular borrelidin PKS, a DH domain creates a typical *E*-olefin, which is later isomerised[37], while in the case of FR901464 a specialised TE domain rather than a DH domain, creates the *Z*-olefin[38]. In the case of the modular fostriecin PKS the DH from module 2 has been shown to create a *Z*-olefin directly[39]. Xie and Cane recently showed that in the cases of bongkrekic acid and oxazolomycin, produced by bacterial *trans*-AT PKS, a KR sets up a β-alcohol anti to an α-proton and the subsequent *syn* dehydration gives the *Z*-olefin directly[40]. The only KR/DH pair from a fungal hr-PKS investigated in vitro is from the SQTKS and this has a KR, which reduces to give the opposite alcohol diastereomer and the subsequent *syn* DH yields the *E*-olefin[41]. The strobilurin PKS is unique in being an iterative type I system that creates the *Z*-olefin. Our results suggest that the strobilurin PKS installs this geometry without the requirement for any

**Fig. 8** Proposed mechanism of the strobilurin rearrangement and (boxed) comparison with a proposed cationic rearrangement mechanism of 2R-littorine. R = 3-tropanyl

other proteins, so the KR/DH domains may be able to control different stereoselectivity during their first (E), second (Z) and third (E) iterations, perhaps in response to methylation. Alternatively, the unknown domain of the strobilurin PKS may have a role in formation of the unusual E,Z,E triene, but further detailed in vitro work will be required to determine its origin.

The released polyketide **11** requires oxidation and rearrangement to form the key acrylate moiety (Fig. 7a). Similar rearrangements have been suggested in the cases of aspyrone **23**[42] and bartanol **24**[43] (Fig. 7b), and are also involved in ring-contracting mechanisms in the case of xenovulene A[44,45]. Tang and co-workers very recently reported a family of CrtC-type (carotenoid 1,2-hydratase) enzymes, which can catalyse rearrangements of epoxides. In the case of the CrtC-enzyme PenF this rearrangement is a Meinwald rearrangement on the pathway to penigequinolone. In our hands expression of the FAD-dependent monooxygenase encoded by str9 in the presence of prestrobilurin A **11** effectively accomplishes the oxidative rearrangement in vivo involving the highly unusual migration of a carboxylate. This reaction is also observed in vitro. We propose that epoxidation of the 2,3 olefin of **11**, followed by Meinwald rearrangement of **25** would furnish the aldehyde intermediate **26**. This intermediate, however, seems to be highly reactive and in the absence of other enzymes it appears to undergo rapid retro-Claisen reaction to give the observed carboxylic acid **21** (Fig. 7a).

In *Strobilurus* species rapid enolisation of **26** would give the β-methoxyacrylate skeleton **27** and methylation (catalysed by Str2 and Str3) would give strobilurin A **1** directly. Further support for the intermediacy of **26** comes from the reductive pathway to **22** and bolineol **8**. The reduction occurs both in the

presence of Stl2 (expt 11, expt 18) and, in *A. oryzae*, absence of Stl2 (expt 17) likely due, at least in part, to a shunt pathway in *A. oryzae*, which is known to reduce aldehydes easily[46]. The FDO Str4 is a member of the GMCO superfamily, which are enzymes responsible for the oxidation of primary alcohols to aldehydes. In the presence of Str4 and absence of the reductase Stl2, the pathway produces strobilurin A **1** and the shunt **21** (expt 12). In the absence of Str4, and the presence of Stl2, the pathway produces bolineol **8** instead (expt 18). This suggests that in *S. tenacellus* the pathway may be controlled to produce **1** or **8** selectively, possibly by control of the individual promoters. In our *A. oryzae* expression more crude control of the promoters (e.g. expt 11) produces both compounds. A possible pathway from **8** to **1** via alcohol oxidation and methylation is also possible, but we could find no LCMS evidence in support of the required mono-methyl intermediate.

The retro-Claisen removal of the β-aldehyde of **26** links this step to the proposed oxidative ring contractions involved during xenovulene A biosynthesis that are also catalysed by FAD-dependent enzymes[45]. However, bioinformatic analysis shows no significant homology between Str9 and AsR4 and AsR6 that are responsible for these reactions in *Sarocladium schorii*[45]. There is also no significant homology between Str9, or any other protein encoded by the *str* BGC, and PenF, suggesting that Str9 accomplishes both the oxidation of **11** and the rearrangement to form **26**.

The proposed rearrangement mechanism involves the unusual migration of a carboxylate as proven by the isotopic labelling studies[10]. Similar processes are known, for example, during the mechanism of methylmalonate mutase and during the biosynthesis of the tropane alkaloid hyoscyamine from littorine **28** (Fig. 8). In the classic case of methylmalonate mutase the reaction is known to be adenosylcobalomin-dependent, and to proceed via single-electron species[47]. In the case of hyoscyamine biosynthesis a cytochrome P450 enzyme is known to catalyse the key rearrangement of littorine **28** to give 2S-hyoscyamine aldehyde **29**. Calculations reported by Sandala et al.[48] suggest that the lowest-energy pathway for this conversion involves rearrangement of a cationic species such as **30** obtained by facile oxidation of an intermediate radical **31**, with acceleration promoted by partial deprotonation of the alcohol and partial protonation of the carboxyl. However, in vitro studies of the enzyme by O'Hagan and co-workers[49] supported a rearrangement mechanism with more radical character. In the strobilurin rearrangement formation of radical intermediates seems unlikely and a cationic mechanism via e.g. **32** is consistent with other Meinwald processes[50] and the calculations for the littorine **28** rearrangement to which it shows remarkable structural similarities (Fig. 8). However, further in vitro experiments will be required to definitively determine the mechanism.

Both *S. lutea* and *S. tenacellus* produce a range of modified strobilurins, including the chlorinated strobilurin B **6**. However, the cluster encodes no obvious halogenase gene that could be involved in its production. Similarly, no obvious dimethylallyl-transferase appears to be encoded within the currently determined BGC. This leaves the biosynthesis of compounds such as strobilurin G **7** as cryptic for the time being. It is possible that unknown proteins encoded in, or near, the BGC (e.g. *str1* and *stl1*) may form new classes of halogenases or dimethylally-transferases, or that the responsible genes are located elsewhere on the genome. Similarly, proteins encoded by *str5/str6* (hydrolases) appear to have no chemical role in the biosynthesis of **1**. No obvious self-resistance gene was found within the cluster and this correlates with previous studies suggesting that the mitochondrial

cytochrome $bc_1$ complex of *S. tenacellus* is not inhibited by β-methoxyacrylates[14] (see Supplementary Methods and Supplementary Figures 45–47 for analysis of the resistance mechanism of *S. lutea* and *S. tenacellus*).

Overall, investigation of the strobilurin biosynthetic pathway in *Strobilurus* species has revealed how nature has generated this valuable class of fungicides, making use of: a previously unobserved biosynthetic pathway to benzoyl CoA **16**; a class of highly reducing iterative fungal PKS, which appears to include a very unusual partial hydrolase and *C*-MeT domains located downstream of the usually terminal ACP, and which is capable of forming a highly unusual *E,Z,E* triene; and a previously unobserved FAD-dependent oxygenase, which is responsible for the key oxidative rearrangement of a polyketide skeleton to form the crucial β-methoxyacrylate toxophore of the strobilurin fungicides. The use of heterologous expression once again illustrates the power of this method for the investigation and engineering of fungal BGCs. In our hands the unoptimised heterologous expression host produced over 100 mg L$^{-1}$ of strobilurin-related metabolites compared to 30 mg L$^{-1}$ in optimised WT fermentations.

## Methods

**Fermentation and extraction**. *S. tenacellus* (strain CBS 621.79) and *B/S. lutea* F23523 were obtained from CBS-KNAW, Fungal Diversity Centre, Netherlands, and Novartis, Switzerland, respectively. A plug of fungi (*S. tenacellus* or *S. lutea* F23523) was inoculated into 100 mL malt extract broth in 500 mL Erlenmeyer flasks and incubated at 200 rpm and 25 °C. After 10 days, the culture was homogenised by a hand blender and used as seed culture. The seed (OD$_{600}$ 0.3) was inoculated in 100 mL of liquid medium (CGC or CMP) and incubated for the desired period. Each culture was then homogenised by hand blender and twice extracted with ethyl acetate with ratio = 1:1 v/v. The organic extracts were pooled, dried (MgSO$_4$), filtered and evaporated in vacuo. The residue was dissolved in methanol and defatted using *n*-hexane (two times) then evaporated in vacuo. Finally, the defatted residue was dissolved in high-performance liquid chromatography (HPLC)-grade methanol (10 mg mL$^{-1}$) and analysed by LCMS.

*A. oryzae* NSAR1 and selected transformants were grown on malt extract agar (MEA) until sporulation (5–7 days), after which the spores were harvested in 5 mL sterilised H$_2$O and used to inoculate production medium (CMP, 100 mL). Cultures were grown for 6 days at 28 °C with shaking at 200 rpm before being homogenised by a hand blender and extracted twice with ethyl acetate (1:1 v/v). The organic layer was separated, dried (MgSO$_4$), filtered and evaporated in vacuo. The extract was dissolved in methanol and defatted using *n*-hexane (two times) then evaporated in vacuo. Finally, the extract was dissolved in HPLC-grade methanol (10 mg mL$^{-1}$) and analysed by LCMS. Samples for purification were prepared to 100 mg mL$^{-1}$.

**Media**. All media were prepared in deionised water and autoclaved at 126 °C for 20 min. CMP, 3.5% Czapek Dox, 2% maltose and 1% peptone; CGC, 50 g L$^{-1}$ glucose, 5 g L$^{-1}$ corn steep liquor and 2 g L$^{-1}$ CaCO$_3$; GN, 1% glucose and 2% nutrient broth No. 2; CZST, 3.5% Czapek Dox agar and 1 M sorbitol; CZSB, 3.5% Czapek Dox broth, 1 M sorbitol and 0.8% agar; SMURA, 0.17% yeast nitrogen base, 0.5% ammonium sulphate, 2% glucose, 0.077% complete supplement mixture minus uracil and 1.5% agar; YM, 4 g L$^{-1}$ yeast extract, 4 g L$^{-1}$ glucose and 10 g L$^{-1}$ malt extract; LB, 1% tryptone, 0.5% yeast extract and 0.5% NaCl; YPAD, 1% (w/v) yeast extract, 2% bactotryptone, 2% (w/v) glucose and 0.04% (w/v) adenine sulphate; MEA, 15 g L$^{-1}$ malt extract, 1.5 g L$^{-1}$ arginine, 1.5 g L$^{-1}$ methionine, 0.1 g L$^{-1}$ adenine, 2 g L$^{-1}$ ammonium sulphate and 15 g L$^{-1}$ agar.

**Analytical LCMS**. LCMS data were obtained with either (LCMS method 1): a Waters 2795HT HPLC a Phenomenex Kinetex column (2.6$\mu$, C$_{18}$, 100 Å, 4.6 × 100 mm) equipped with a Phenomenex Security Guard precolumn (Luna C$_5$ 300 Å) eluted at 0.9 mL min$^{-1}$, with a Waters 996 Diode Array detector between 200 and 600 nm and a Waters ZQ mass detector operating simultaneously in ES$^+$ and ES$^-$ modes between 100 and 650 *m/z*; or (LCMS method 2) a Waters 2767 sample manager connected to Waters 2545 pumps and SFO, a Phenomenex Kinetex column (2.6$\mu$, C$_{18}$, 100 Å, 4.6 × 100 mm) equipped with a Phenomenex Security Guard precolumn (Luna C$_5$ 300 Å) eluted at 1.0 mL min$^{-1}$, with a waters 2998 Diode Array detector (200–600 nm) and Waters 2424 ELSD and Waters SQD-2 mass detector operating simultaneously in ES$^+$ and ES$^-$ modes between 100 and 650 *m/z*. Solvents were: A, HPLC-grade H$_2$O containing 0.05% formic acid; B, HPLC-grade MeOH containing 0.045% formic acid; and C, HPLC-grade CH$_3$CN containing 0.045% formic acid. The gradient was as

follows: 0 min, 10% C; 10 min, 90% C; 12 min, 90% C; 13 min, 10% C; and 15 min, 10% C.

**Compound purification**. Purification of all compounds was generally achieved using a Waters mass-directed autopurification system comprising of a Waters 2767 autosampler, Waters 2545 pump system, a Phenomenex Kinetex Axia column (5$\mu$, C$_{18}$, 100 Å, 21.2 × 250 mm) equipped with a Phenomenex Security Guard precolumn (Luna C$_5$ 300 Å) eluted at 20 mL min$^{-1}$ at ambient temperature. Solvents as above. The post-column flow was split (100:1) and the minority flow was made up with HPLC-grade MeOH + 0.045% formic acid to 1 mL min$^{-1}$ for simultaneous analysis by diode array (Waters 2998), evaporative light-scattering (Waters 2424) and electrospray ionisation mass spectrometry in positive and negative modes (Waters SQD-2). Detected peaks were collected into glass test tubes. Combined fractions were evaporated (vacuum centrifuge), weighed and residues dissolved directly in deuterated solvent for NMR.

**Construction of expression plasmids**. All construction details are given in the Supplementary Methods.

**Transformation of *A. oryzae* NSAR1**. Spores of *A. oryzae* NSAR1 were prepared by inoculation of the fungus onto MEA plates and incubation at 30 °C for 1–2 weeks or until sporulation occurred. The spores were harvested from plates and inoculated into GN medium then incubated at 28 °C with shaking 200 rpm overnight. The culture was harvested by centrifugation at 8000 × *g* for 10 min and the supernatant discarded. The pellet was washed once with sterilised water and once with sterile 0.8 M NaCl. The pellet was resuspended in 10 mL of filter sterilised protoplasting solution (20 mg mL$^{-1}$ trichoderma lysing enzyme and 5 mg mL$^{-1}$ driselase in 0.8 M NaCl) and incubated at room temperature with rotary shaking. After 1–1.5 h, protoplasts were released from hyphae by pipetting with a wide-bore tip then filtered through two layers of sterile miracloth. The filtrate was centrifuged at 1000 × *g* for 5 min and the supernatant discarded. The pellet was washed with solution 1 (0.8 M NaCl, 10 mM CaCl$_2$ and 50 mM Tris-HCl, pH 7.5). The pellet was resuspended in 200–500 μL of solution 1. Then, 100 μL of protoplast was mixed gently with 5–10 μg (10 μL) of plasmid DNA and incubated on ice 2 min. The mixture was added 1 mL of solution 2 (60% (w/v) PEG 3350, 0.8 M NaCl, 10 mM CaCl$_2$ and 50 mM Tris-HCl, pH 7.5) and incubated for 20 min. Finally, the mixture was added 40 mL of molten (50 °C) CZSB and overlaid onto four prepared plates containing 15 mL of CZST plus appropriate additives (see below). The plates were incubated at 28 °C for 3–5 days until colonies appeared.

**Selective media for *A. oryzae* transformation**. A volume of 100 mL of 3.5% Czapek Dox agar was supplemented with 1 mL of 20% ammonium sulphate and 1 mL of an optional supplement (depending on selection marker of plasmid) such as 0.5% adenine and/or 1% methionine and/or 2% arginine.

**Nuclear magnetic resonance**. NMR was obtained using a Bruker Avance 500 instrument equipped with a cryo-cooled probe at 500 MHz ($^1$H) and 125 MHz ($^{13}$C). 2D spectra (COSY, HSQC and HMBC) were obtained using standard parameters. Samples were dissolved in the indicated solvents. $^1$H and $^{13}$C spectra are referencd relative to residual protonated solvent. All δ values are quoted in ppm and all *J* values in Hz.

## Data availability

All NMR data, details of cloning procedures, detailed LCMS chromatograms and details of bioinformatic procedures and results are contained in the Supplementary Information. The strobilurin BGC is deposited at GenBank with accession number KY070339. All other data are available from the authors upon reasonable request.

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

## Acknowledgements

R.N. thanks the Directorate General of Resources for Science and Higher Education (Beasiswa Pendidikan Pascasarjana Luar Negeri Direktorat Jenderal Sumber Daya Ilmu Pengetahuan dan Pendidikan Tinggi (BPP-LN Ditjen SD Iptek-Dikti)), Republic of Indonesia, for an Overseas Postgraduate Scholarship and the Alumni foundation of the University of Bristol for financial support. K.d.M.-S. was funded by BBSRC (BB/K002341/1) and L.-C.H. by BBSRC and EPSRC through BrisSynBio and the Centre for Synthetic Biology (BB/L01386X/1). Z.I. thanks the Higher Education Commission of Pakistan for a post-graduate scholarship. We thank EPSRC (EP/F066104/1) and DFG (INST 187/621–1) for LCMS equipment, BBSRC for culture facilities (BB/L01386X/1) and the University of Bristol Genomics Facility for DNA and RNA sequencing. K.E.L. was funded by the Leibniz University of Hannover. R.J.C. thanks Dr. John Clough and

Dr. Katherine Williams for helpful discussions, and Jennifer Senkler (Leibniz Universität) for protein MS analysis.

## Author contributions

R.J.C. and T.J.S. designed the study and wrote the manuscript. R.N. and Z.I. fermented the fungi and performed the natural product analysis. A.M.B., R.N. and K.d.M.-S. performed the genome and transcriptome sequence, analysis and interpretation. R.N., K.d. M.S. and K.E.L. performed the cloning and heterologous expression. R.N. and K.E.L. performed the compound isolation, analysis and structure elucidation. L.-C.H. performed all synthetic procedures. L.C.H., Z.I., R.N. and K.E.L. were supervised by R.J.C., T.J.S., C.L.W. and A.M.B. All authors revised the MS.

## Additional information

**Competing interests:** The authors declare no competing interests.

