## [Peer Review File · Nature Communications]

Reviewers' comments:

Reviewer #1 (Remarks to the Author):

This work described the biosynthesis mechanism for strobilurins in basidiomycete fungi. Strobilurins are a valuable class of fungicides used worldwide to treat a broad range of plant pathogens. Although they are simple small molecules, their biosynthetic mechanism had not been fully understood. There are many challenges to elucidate the molecular mechanism for biosynthesis of natural products from basidiomycete fungi. One clear challenge is the numerous introns in these basidiomycete genes. So the correct gene annotation through transcriptome sequencing, successful amplification of the large intron-free PKS gene, functional expression of various combinations of the gene cluster in a heterologous host, decent-yield production and clear identification of the resulted natural products are impressive. These efforts verified the biosynthetic gene cluster for strobilurins, identified biosynthetic intermediates, and established the sequence of individual biosynthetic steps.

There are several interesting features in the biosynthesis of the β -methoxyacrylate fungicides. The most interesting aspect is the unusual oxidative rearrangement involved in the formation of the core β -methoxyacrylate. However, no experimental effort appeared to tackle this aspect. The authors are suggested to carry out experiments to address this most interesting point, such as isotope labelling to confirm the proposed rearrangement. For example, if the proposed mechanism is correct, NMR would show C1 and C2 of compound 1 are from two different acetate units, and the methoxy carbon and C1 are from the same acetate unit. The results would make the manuscript significantly stronger.

Minor comments:

1. Although not directly related to the main topic of the work, is it a common practice to reclassify a previously established fungal species solely based on the sequence analysis of its internal transcribed spacer (ITS)? The authors are suggested to include references or other supporting information for the rename of *Bolinea lutea* to *Strobilurus lutea*.
2. As the authors pointed out, the strobilurin PKS is unusual among fungal hr-PKS in containing catalytic domains located after the ACP domain. Sequence analysis suggested that the C-terminal C-MeT domain is likely to be active, and this domain was proposed to attach the C-4 methyl group. Are the domains really required for the biosynthesis of strobilurins? Could any of the domains be related to product release, which remains unknown? Simple point mutation experiments could help clarify these questions.

Reviewer #2 (Remarks to the Author):

This study by the Cox group identified and characterised the biosynthetic gene cluster for strobilurin A by heterologous expression in *Aspergillus oryzae*. The study has some significance and novelties:

- 1) Strobilurins are antifungal compounds produced by several basidiomycete fungi. They are the basis for the development of the QoI fungicide class that covers majority of the newest and most important fungal disease-control chemicals in agriculture. Many synthetic derivatives have been developed, yet, the molecular basis for its biosynthesis was not known prior to this study.
- 2) Reconstruction of a basidiomycete biosynthetic pathway in an ascomycete host represent a technological feat and set the basis for exploration of other basidiomycete pathways.

3) The PKS that synthesise the backbone has unique architecture

4) The str9 has been shown by heterologous expression to catalyse an interesting epoxidation-mediated Meinwald rearrangement.

However, for it to be published in a Nature series journal, one would expect more. There are several weaknesses below:

1) There is considerable description and discussion on the uniqueness of SIPKS1/StPKS1. However, little is done for characterising the PKS. Given that the CMeT is at the C-terminal and the original CMeT was claimed to be no longer functional, is the methylation added post extension or during extension? Could the chain extension proceed and produced a demethylated version of 11 when the C-terminal CMeT domain is deleted?

2) The authors did a great job characterising the benzoyl-CoA biosynthesis, but when it comes to the later steps of the strobilurin biosynthesis, it was not performed to the same level of rigorousness. For example, experiments omitting only the GMCO or only one of the two O-methyltransferases were not performed (Table 3). Leaving the open questions of the actual function of GMCO and the exact role/timing of the two O-methyltransferases. Given the known function of GMCO (e.g. choline dehydrogenase) and no 22 is observed in expt11 and 12 (Table 3), is it possible that the enzyme catalyse the conversion of 22 back to 26, effectively redirecting shunt product back to the pathway to 1? Which OMeT catalyse the formation of methyl ester (not very common)?

3) The Str9 is a major highlight in this study. But the characterisation is limited to heterologous expression in *A. oryzae*. Given that the substrate 11 is available, what is stopping the authors from characterising the biochemical reaction in vitro?

Other comments (based on the order appeared in the text):

1) Table 1 is confusing. The two columns looks like a continuing list of genes at first glance, it should add a divider and another row on the top indicating the fungal species where the genes came from, and their homology as well.

2) How was slr4 assigned as a GMCO and not just a FDO is unclear, since both are FAD dependent?

3) The first paragraph in page 6 should be continued/merged with the previous paragraph, as it is still going on about the PKS. Starting a paragraph with "This sequence is..." in the absent of context confuses the reader.

4) The synthesis of 11 section seems redundant. Can be moved to SI. Since the production of 11 is 20mg/L, it should be more than sufficient for NMR, as it is shown. Not sure about the aim of the synthesis.

5) Last paragraph in page 8, '(CoA ligase)' should be right after when 'str10' is first appeared in the paragraph not later. Same for str8 (non-heme iron oxygenase). If it was already described earlier in the same paragraph, there is no need for it to appear again and again in the same paragraph.

6) Figure 7, suggest include the side product from the retro Claisen reaction.

7) Page 10 paragraph on top, "In a second experiment led to the same result." – it is not the same! Expt 11 has 11 plus multiple * peaks considered as unrelated. It needs better explanation.

8) Figure 8, A-G could directly referred to the expt. no. in Table 3.

9) Page 11, please toned down the novelty on "...the strobilurin PKS is highly unusual among fungal hr-PKS in containing catalytic domains located after the ACP domain. The presence of non-canonical catalytic domains such as C-MeT and reductive release domains is common for non-reducing PKS (e.g. methylorcinolaldehyde synthase), but currently unknown for other classes of fungal PKS." This is not true! All PKS-NRPS hybrids are essentially HR-PKS with C-A-T-R catalytic domains after the ACP domain. There are also the HR-PKS with a C-terminal carnitine-AT characterised recently by the Tang group (please cite).

Reviewer #3 (Remarks to the Author):

The authors uncovered the biosynthetic pathway for the strobilurin class of antifungal polyketides through a multidisciplinary set of experiments. By sequencing two producer strains, genome mining them for relevant polyketide and starter-unit biosynthesis genes, heterologously expressing selected sets of pathway genes, and chemically synthesizing product standards, the authors were able to characterize the biosynthetic pathway for strobilurin A and demonstrate the enzymes responsible for the benzoyl starter unit and, more impactfully, the β -methoxyacrylate toxophore moiety. The experiments were well-executed and the conclusions well-founded. The proposed enzymatic steps for strobilurin biosynthesis were nicely described and discussed in the context of analogous enzymes from the scientific literature, particularly for the Meinwald rearrangement, and should be of general interest to the scientific community.

However, there are some weak points with the article that should be addressed. Overall, these weaknesses are minor/cosmetic and could be taken care of with mainly superficial edits.

1) The authors should thoroughly proofread both the main text and supporting information for grammatical errors. At times the tense seemed inappropriate (mainly in the abstract). Relative clauses (which/that) were very often used incorrectly. Commas were often misplaced or lacking. Such grammatical improvements would increase clarity.

2) Check that abbreviations have been defined once, then used consistently. For example, ATP, CoA, SDR, and LCMS were never defined in the main text, whereas many PKS domain names were defined on several occasions and inconsistently used. Retention time was abbreviated to RT or Rt; choose one and be consistent.

3) Figure subsections were inconsistent, sometimes labeled with uppercase letters, sometimes lowercase. I believe Nature Communications uses lowercase in bold lettering.

4) Recommended to use full gene names, e.g. slr11, rather than shortening to r11, for clarity throughout the text.

5) The introduction should have a brief description of how PKSs work, since Nature Communications aims for a general audience and this is relevant to understanding the main text.

6) Figure 1; It would be helpful to indicate in the caption the source of the compounds as natural or synthetic. This way the reader can quickly assess which structures are biosynthetically relevant. E.g. "... key natural (1-2, 5-8) and synthetic (3-4) strobilurins." Also, the kresoxim methyl 4 toxophore is a methoxyiminoacetate, not β -methoxyacrylate, as indicated in the caption.

7) Sometimes a strong conclusion statement is made in the main text without enough information about its basis: e.g., from pg. 3, "The data also clearly indicated that the organism previously known as *Bolinea lutea* is, in fact, a basidiomycete, and sequence analysis of its internal transcribed spacer (ITS) sequence showed it to be a previously unreported member of the *Strobilurus* family (Supplementary Figure 8)." This phylogenetic tree figure was created with a very limited set of a single marker gene region (ITS) sequences and no bootstrap values were given to tell the confidence of the groupings. I recommend softening the language to either the ITS data "is consistent with" or "likely indicates" *Bolinea lutea* is a Basidiomycete that groups with *Strobilurus* strains. If any related *Bolinea lutea* ITS sequences are available, they should be included in the tree to show that the new sequence clusters differently. Fungal class names should be capitalized, e.g. Basidiomycete.

8) Pg.4; I believe up and downstream were inverted in this sentence: "We therefore set the approximate cluster boundaries as *stl2* upstream of the PKS and *str11* downstream of the PKS, defining a ca 47 kb sequence for more detailed analysis."

9) Figure 3; coloring of Redox and MeT genes may be difficult to differentiate.

10) Pg.6 top; recommend to refer to TE region in *stPKS1* as a "partial TE" to avoid confusion.

11) Pg. 6; please give accession numbers for *Stereum hirsutum* and *Gloeophllum trabeum* proteins that align to the unknown domain in the main text, since a paper reference can't be provided as for the other homologs.

12) Inconsistent use of lowercase k vs. uppercase K to indicate kilo (kb vs. KDa). I believe kilo should be with a lowercase k, as in kDa.

13) Figure 4; off-set plot from baseline, so visible, and indicate Y-axis wavelength for DAD (also define DAD term)

14) Figure 5; needs an expanded caption to reference that 2D NMR data is shown and that benzoyl-SNAC is accepted as a starter-unit, but free benzoic acid is not.

15) Pg. 8; rather than somewhat ambiguous "PKS-free" term, consider using isolable or unbound.

16) Table 2; The caption should describe the experimental setup more clearly. A line could be added to top of the table to indicate which components are the genes co-expressed, the substrates supplemented to the culture, and the product detected.

17) Figure 7; bottom line could have an additional arrow from 16 catalyzed by *stPKS1* to 11 (just the number without the full structure) since it's included in the co-expressions and is the reporter product relevant to this figure.

18) Figure 8; off-set chromatogram from baseline so visible. Colors aren't necessary. Indicate Y-axis units (are these on the same scale?).

19) Pg. 11; In discussing the benzoate starter unit, the possibility for self-loading by the ACP or a potential role of the unknown domain (between the partial TE and the C-terminal C-MeT) in transferring the starter unit is not given as an alternative to AT-selection of benzoate. Have you considered these possibilities and if so, why did you rule them out?

20) Figure 9; give compound numbers for aspyrone 23 and bartanol 24 in caption.

21) Pg. 13; both *Strobilurus* species and strobilurus species are used. I believe the first is correct.

22) pg. 13; This section has a few issues: "This facile shunt in *A. oryzae* means that it [is] not possible to determine whether the SDR is capable of catalysing this step in strobilurus species, but the results do show that the methyltransferases cannot act on the reduced bolineol skeleton and are therefore selective for the β -methoxyacrylate. It may be that bolineol 8 is therefore a reduction product of 1 in strobilurus species, however if that is the case, the SDR is not catalysing this reaction as 8 it is not observed in the *A. oryzae* expression (Table 3 expt 11, Figure 8A)." Insert "is" at the location shown in brackets. As written, β -methoxyacrylate is the proposed substrate of the methyltransferases, instead of the product. Please rephrase to indicate that the MTs are proposed to act on the enolized β -hydroxyacrylate 27. It would be helpful to give the protein names in parentheses for clarity when mentioning the SDR, i.e. ...the SDR (stl2)....

23) Pg. 13; in discussing the origin of bolineol, have you considered the possibility that the reductase could act after a single methyl transfer to the carboxyl moiety to give bolineol as an alternative product, but that in coexpressions (which always had both MT genes) the second MT outcompetes reduction to give only product 1?

24) Pg 16; de-fatted vs. defatted and g/L vs. g L⁻¹, select one spelling/nomenclature and be consistent. First line, second paragraph; were spores harvested from malt extract liquid culture (as written) or agar plates? In Media section: 0.077% (replace comma with period and delete space), YPAD, 1% (no space between number and percent sign). Use hyphen to indicate a range of numbers.

25) Pg. 17; In analytical LCMS section. Capitalize Waters.

I've also attached a pdf of the supporting information with suggested edits as comments.

Response to Reviewers' Comments

Responses in red

Reviewer #1 (Remarks to the Author):

This work described the biosynthesis mechanism for strobilurins in basidiomycete fungi. Strobilurins are a valuable class of fungicides used worldwide to treat a broad range of plant pathogens. Although they are simple small molecules, their biosynthetic mechanism had not been fully understood. There are many challenges to elucidate the molecular mechanism for biosynthesis of natural products from basidiomycete fungi. One clear challenge is the numerous introns in these basidiomycete genes. So the correct gene annotation through transcriptome sequencing, successful amplification of the large intron-free PKS gene, functional expression of various combinations of the gene cluster in a heterologous host, decent-yield production and clear identification of the resulted natural products are impressive. These efforts verified the biosynthetic gene cluster for strobilurins, identified biosynthetic intermediates, and established the sequence of individual biosynthetic steps.

There are several interesting features in the biosynthesis of the β -methoxyacrylate fungicides. The most interesting aspect is the unusual oxidative rearrangement involved in the formation of the core β -methoxyacrylate.

However, no experimental effort appeared to tackle this aspect. The authors are suggested to carry out experiments to address this most interesting point, such as isotope labelling to confirm the proposed rearrangement. For example, if the proposed mechanism is correct, NMR would show C1 and C2 of compound 1 are from two different acetate units, and the methoxy carbon and C1 are from the same acetate unit. The results would make the manuscript significantly stronger.

This work has already been performed and previously reported in the literature, e.g. see refs 10 and 11.

Minor comments:

1. Although not directly related to the main topic of the work, is it a common practice to reclassify a previously established fungal species solely based on the sequence analysis of its internal transcribed spacer (ITS)? The authors are suggested to include references or other supporting information for the rename of *Bolinea lutea* to *Strobilurus lutea*.

In fact the ITS reclassification prefigured analysis of the entire genome. It is quite clear that the fungus is unrelated to other *Bolinea* species and has been incorrectly classified. Its closest neighbors are *Strobilurus* species. The ITS analysis is merely a convenient - and widely accepted - way of showing this. Analysis of any other gene from the sequenced genome can be used to draw the same conclusion. This highlights the difficulty of determining correct classification in fungi without genomic data.

2. As the authors pointed out, the strobilurin PKS is unusual among fungal hr-PKS in containing catalytic domains located after the ACP domain. Sequence analysis suggested that the C-terminal C-MeT domain is likely to be active, and this domain was proposed to attach the C-4 methyl group. Are the domains really required for the biosynthesis of strobilurins? Could any of the domains be related to product release, which remains unknown? Simple point mutation experiments could help clarify these questions.

These are good questions, but would take a long time to answer. We wish to publish our current observations in *Nature Communications* so that a wide audience can rapidly access this data. Future work by us or others will no doubt begin to unpick these interesting questions building upon our current results.

Reviewer #2 (Remarks to the Author):

This study by the Cox group identified and characterised the biosynthetic gene cluster for strobilurin A by heterologous expression in *Aspergillus oryzae*. The study has some significance and novelties:

1) Strobilurins are antifungal compounds produced by several basidiomycete fungi. They are the basis for the development of the QoI fungicide class that covers majority of the newest and most important fungal disease-control chemicals in agriculture. Many synthetic derivatives have been developed, yet, the molecular basis for its biosynthesis was not known prior to this study.

- 2) Reconstruction of a basidiomycete biosynthetic pathway in an ascomycete host represent a technological feat and set the basis for exploration of other basidiomycete pathways.
- 3) The PKS that synthesise the backbone has unique architecture
- 4) The str9 has been shown by heterologous expression to catalyse an interesting epoxidation-mediated Meinwald rearrangement.

However, for it to be published in a Nature series journal, one would expect more. There are several weaknesses below:

1) There is considerable description and discussion on the uniqueness of SIPKS1/StPKS1. However, little is done for characterising the PKS. Given that the CMeT is at the C-terminal and the original CMeT was claimed to be no longer functional, is the methylation added post extension or during extension? Could the chain extension proceed and produced a demethylated version of 11 when the C-terminal CMeT domain is deleted?

As above, these are interesting questions, but are *tangential* to the main thrust of the current manuscript, which is to elucidate the pathway and BGC, and would take a very long time to answer convincingly. No doubt we or others will do this, but for rapid dissemination to a wide audience we wish to report our new observations now. Extra text is now included to summarise the basis for these assertions, and detailed sequence alignment is shown in the supplementary information.

2) The authors did a great job characterising the benzoyl-CoA biosynthesis, but when it comes to the later steps of the strobilurin biosynthesis, it was not performed to the same level of rigorousness. For example, experiments omitting only the GMCO or only one of the two O-methyltransferases were not performed (Table 3). Leaving the open questions of the actual function of GMCO and the exact role/timing of the two O-methyltransferases. Given the known function of GMCO (e.g. choline dehydrogenase) and no 22 is observed in expt11 and 12 (Table 3), is it possible that the enzyme catalyse the conversion of 22 back to 26, effectively redirecting shunt product back to the pathway to 1?

We have now completed experiments defining the roles of the GMCO and SDR. New data and further discussion is included in the MS and evidence is now provided to support this very point.

Which OMeT catalyse the formation of methyl ester (not very common)?

Again this is tangential to the main story of the MS. Further work will inevitably solve this question, but for now it would take at least another 3 months of work and not significantly increase the interest of impact of the main observations.

3) The Str9 is a major highlight in this study. But the characterisation is limited to heterologous expression in *A. oryzae*. Given that the substrate 11 is available, what is stopping the authors from characterising the biochemical reaction *in vitro*?

The problem has been the insolubility of 11 in aqueous solution. We have purified Str9 from *E. coli* expression, but early assays showed no activity and we thus did not include the work in the first draft of the MS. However, recently we have found conditions under which limited turn-over of 11 is observed *in vitro*. This experiment is now referred to in the MS, and the data is included in the supplementary information where extracted ion chromatogram data are shown. Currently we are unable to obtain soluble GMCO so further *in vitro* work cannot yet be done.

Other comments (based on the order appeared in the text):

1) Table 1 is confusing. The two columns looks like a continuing list of genes at first glance, it should add a divider and another row on the top indicating the fungal species where the genes came from, and their homology as well.

This has been done.

2) How was slr4 assigned as a GMCO and not just a FDO is unclear, since both are FAD dependent?

NCBI conserved domain analysis shows this protein to be a member of the GMC oxidoreductase (GMCO) superfamily. Its closest verified relative is aryl alcohol dehydrogenase, involved in the production of peroxide in *Pleurotus* species.

3) The first paragraph in page 6 should be continued/merged with the previous paragraph, as it is still going on about the PKS. Starting a paragraph with "This sequence is..." in the absent of context confuses the reader.

This has been done.

4) The synthesis of 11 section seems redundant. Can be moved to SI. Since the production of 11 is 20mg/L, it should be more than sufficient for NMR, as it is shown. Not sure about the aim of the synthesis.

The synthesis gives additional evidence of the *EZE* stereochemistry, and provides the ultimate level of confidence in the structural assignment.

5) Last paragraph in page 8, '(CoA ligase)' should be right after when 'str10' is first appeared in the paragraph not later. Same for str8 (non-heme iron oxygenase). If it was already described earlier in the same paragraph, there is no need for it to appear again and again in the same paragraph.

The structure of the sentence explicitly makes these clear the first time. I have deleted the later instances.

6) Figure 7, suggest include the side product from the retro Claisen reaction.

This has been done.

7) Page 10 paragraph on top, "In a second experiment led to the same result." – it is not the same! Expt 11 has 11 plus multiple * peaks considered as unrelated. It needs better explanation.

The wording has been changed to clarify this point.

8) Figure 8, A-G could directly referred to the expt. no. in Table 3.

This has been done

9) Page 11, please toned down the novelty on "...the strobilurin PKS is highly unusual among fungal hr-PKS in containing catalytic domains located after the ACP domain. The presence of non-canonical catalytic domains such as C-MeT and reductive release domains is common for non-reducing PKS (e.g. methylorcinaldehyde synthase), but currently unknown for other classes of fungal PKS." This is not true! All PKS-NRPS hybrids are essentially HR-PKS with C-A-T-R catalytic domains after the ACP domain. There are also the HR-PKS with a C-terminal carnitine-AT characterised recently by the Tang group (please cite).

This has been done.

Reviewer #3 (Remarks to the Author):

The authors uncovered the biosynthetic pathway for the strobilurin class of antifungal polyketides through a multidisciplinary set of experiments. By sequencing two producer strains, genome mining them for relevant polyketide and starter-unit biosynthesis genes, heterologously expressing selected sets of pathway genes, and chemically synthesizing product standards, the authors were able to characterize the biosynthetic pathway for strobilurin A and demonstrate the enzymes responsible for the benzoyl starter unit and, more impactfully, the beta-methoxyacrylate toxophore moiety. The experiments were well-executed and the conclusions well-founded. The proposed enzymatic steps for strobilurin biosynthesis were nicely described and discussed in the context of analogous enzymes from the scientific literature, particularly for the Meinwald rearrangement, and should be of general interest to the scientific community.

However, there are some weak points with the article that should be addressed. Overall, these weaknesses are minor/cosmetic and could be taken care of with mainly superficial edits.

The authors should thoroughly proofread both the main text and supporting information for grammatical errors. At times the tense seemed inappropriate (mainly in the abstract). Relative clauses (which/that) were very often used incorrectly. Commas were often misplaced or lacking. Such grammatical improvements would increase clarity.

Grammar has been improved throughout with the aid of MS Word. Nature guidelines are that the abstract should be written in the present tense.

2) Check that abbreviations have been defined once, then used consistently. For example, ATP, CoA, SDR, and LCMS were never defined in the main text, whereas many PKS domain names were defined on several occasions and inconsistently used. Retention time was abbreviated to RT or Rt; choose one and be consistent.

This has been done. RT = Room temperature, Rt = retention time.

3) Figure subsections were inconsistent, sometimes labeled with uppercase letters, sometimes lowercase. I believe Nature Communications uses lowercase in bold lettering.

This has been changed.

4) Recommended to use full gene names, e.g. slr11, rather than shortening to r11, for clarity throughout the text.

This has been done.

5) The introduction should have a brief description of how PKSs work, since Nature Communications aims for a general audience and this is relevant to understanding the main text.

This has been done and reference to a primer about fungal PKS systems is given.

6) Figure 1; It would be helpful to indicate in the caption the source of the compounds as natural or synthetic. This way the reader can quickly assess which structures are biosynthetically relevant. E.g. "... key natural (1-2, 5-8) and synthetic (3-4) strobilurins." Also, the kresoxim methyl 4 toxophore is a methoxyiminoacetate, not β -methoxyacrylate, as indicated in the caption.

This has been done.

7) Sometimes a strong conclusion statement is made in the main text without enough information about its basis: e.g., from pg. 3, "The data also clearly indicated that the organism previously known as *Bolinea lutea* is, in fact, a basidiomycete, and sequence analysis of its internal transcribed spacer (ITS) sequence showed it to be a previously unreported member of the *Strobilurus* family (Supplementary Figure 8)." This phylogenetic tree figure was created with a very limited set of a single marker gene region (ITS) sequences and no bootstrap values were given to tell the confidence of the groupings. I recommend softening the language to either the ITS data "is consistent with" or "likely indicates" *Bolinea lutea* is a Basidiomycete that groups with *Strobilurus* strains. If any related *Bolinea lutea* ITS sequences are available, they should be included in the tree to show that the new sequence clusters differently. Fungal class names should be capitalized, e.g. Basidiomycete.

The text has been softened and Bootstrap values have been added to the figure.

8) Pg.4; I believe up and downstream were inverted in this sentence: "We therefore set the approximate cluster boundaries as stl2 upstream of the PKS and str11 downstream of the PKS, defining a ca 47 kb sequence for more detailed analysis."

Thanks!

9) Figure 3; coloring of Redox and MeT genes may be difficult to differentiate.

Red and Magenta have been used successfully in a previous *Nature Communications* paper.

10) Pg.6 top; recommend to refer to TE region in stPKS1 as a “partial TE” to avoid confusion.

I believe the context makes this clear already, but I have clarified on page 17.

11) Pg. 6; please give accession numbers for *Stereum hirsutum* and *Gloephllum trabeum* proteins that align to the unknown domain in the main text, since a paper reference can't be provided as for the other homologs.

These have been added.

12) Inconsistent use of lowercase k vs. uppercase K to indicate kilo (kb vs. KDa). I believe kilo should be with a lowercase k, as in kDa.

This has been changed.

13) Figure 4; off-set plot from baseline, so visible, and indicate Y-axis wavelength for DAD (also define DAD term)

This has been done and DAD has been defined.

14) Figure 5; needs an expanded caption to reference that 2D NMR data is shown and that benzoyl-SNAC is accepted as a starter-unit, but free benzoic acid is not.

Done.

15) Pg. 8; rather than somewhat ambiguous “PKS-free” term, consider using isolable or unbound.

Done.

16) Table 2; The caption should describe the experimental setup more clearly. A line could be added to top of the table to indicate which components are the genes co-expressed, the substrates supplemented to the culture, and the product detected.

More text has been added to the legend. The table headings already clearly differentiate genes (e.g. *str10*) from compounds (e.g. Benzoyl SNAC 10).

17) Figure 7; bottom line could have an additional arrow from 16 catalyzed by stPKS1 to 11 (just the number without the full structure) since it's included in the co-expressions and is the reporter product relevant to this figure.

Done.

18) Figure 8; off-set chromatogram from baseline so visible. Colors aren't necessary. Indicate Y-axis units (are these on the same scale?).

Done. Data is not presented at same scale – because different experiments produced varying amounts of material

19) Pg. 11; In discussing the benzoate starter unit, the possibility for self-loading by the ACP or a potential role of the unknown domain (between the partial TE and the C-terminal C-MeT) in transferring the starter unit is not given as an alternative to AT-selection of benzoate. Have you considered these possibilities and if so, why did you rule them out?

Self-loading is known in Type II systems, particularly for β -carbonyl species, but has not been reported for Type I acyl carrier proteins. It is not impossible, just not particularly likely.

20) Figure 9; give compound numbers for aspyrone 23 and bartanol 24 in caption.

Done.

21) Pg. 13; both Strobilurus species and strobilurus species are used. I believe the first is correct.

Done.

22) pg. 13; This section has a few issues: "This facile shunt in *A. oryzae* means that it [is] not possible to determine whether the SDR is capable of catalysing this step in strobilurus species, but the results do show that the methyltransferases cannot act on the reduced bolineol skeleton and are therefore selective for the β -methoxyacrylate. It may be that bolineol 8 is therefore a reduction product of 1 in strobilurus species, however if that is the case, the SDR is not catalysing this reaction as 8 it is not observed in the *A. oryzae* expression (Table 3 expt 11, Figure 8A)." Insert "is" at the location shown in brackets. As written, β -methoxyacrylate is the proposed substrate of the methyltransferases, instead of the product. Please rephrase to indicate that the MTs are proposed to act on the enolized β -hydroxyacrylate 27. It would be helpful to give the protein names in parentheses for clarity when mentioning the SDR, i.e. ...the SDR (stl2)....

Done.

23) Pg. 13; in discussing the origin of bolineol, have you considered the possibility that the reductase could act after a single methyl transfer to the carboxyl moiety to give bolineol as an alternative product, but that in coexpressions (which always had both MT genes) the second MT outcompetes reduction to give only product 1?

We have not been able to isolate or detect compounds supporting other pathways, although we cannot rule them out completely. Our current preferred route is one in which the GMCO 'rescues' shunted material back into the main pathway which is now supported by more evidence and clarified text in the discussion section.

24) Pg 16; de-fatted vs. defatted and g/L vs. g L⁻¹, select one spelling/nomenclature and be consistent. First line, second paragraph; were spores harvested from malt extract liquid culture (as written) or agar plates? In Media section: 0.077% (replace comma with period and delete space), YPAD, 1% (no space between number and percent sign). Use hyphen to indicate a range of numbers.

Done.

25) Pg. 17; In analytical LCMS section. Capitalize Waters.

Done.

I've also attached a pdf of the supporting information with suggested edits as comments.

These have been addressed.

REVIEWERS' COMMENTS:

Reviewer #1 (Remarks to the Author):

In the revised manuscript, the authors addressed two of the three issues in my review of the previous submission of the manuscript. One is the unusual oxidative rearrangement involved in the formation of the core β -methoxyacrylate moiety essential for biological activity. The authors already reported some results in a separate journal (in press, ref 10). These included isotope labeling studies and intermediate isolation to provide evidence for the involvement of an epoxide in the key rearrangement to form the β -methoxyacrylate. Reporting the results in a separate publication "diluted" current work, but the issue of experimentally exploring the unusual rearrangement has been addressed by the authors. In particular, the authors provided in vitro data in the revision to support that a FAD-dependent oxygenase is responsible for the key oxidative rearrangement of the polyketide skeleton to form the crucial β -methoxyacrylate. In addition, the authors properly addressed the ITS reclassification of *Bolinea* species to *Strobilurus* species.

The authors did not provide new data to address the comment on the unusual features of the strobilurin PKS. It remains unclear whether the unusual domains are required for the strobilurin release or involved in other aspects such as the unusual E,Z,E triene that is formed by the same single-module PKS. However, I understand these would take a long time to answer and the authors wish to publish our current observations quickly.

Reviewer #2 (Remarks to the Author):

The manuscript has been revised based on the reviewers' comments. The authors have adequately answered the questions raised by the reviewers. Together with the additional experimental data included, the quality of the manuscript has improved and is acceptable for publication by Nature Communications.

REVIEWERS' COMMENTS:

Reviewer #1 (Remarks to the Author):

In the revised manuscript, the authors addressed two of the three issues in my review of the previous submission of the manuscript. One is the unusual oxidative rearrangement involved in the formation of the core β -methoxyacrylate moiety essential for biological activity. The authors already reported some results in a separate journal (in press, ref 10). These included isotope labeling studies and intermediate isolation to provide evidence for the involvement of an epoxide in the key rearrangement to form the β -methoxyacrylate. Reporting the results in a separate publication "diluted" current work, but the issue of experimentally exploring the unusual rearrangement has been addressed by the authors. In particular, the authors provided in vitro data in the revision to support that a FAD-dependent oxygenase is responsible for the key oxidative rearrangement of the polyketide skeleton to form the crucial β -methoxyacrylate. In addition, the authors properly addressed the ITS reclassification of *Bolinea* species to *Strobilurus* species.

The authors did not provide new data to address the comment on the unusual features of the strobilurin PKS. It remains unclear whether the unusual domains are required for the strobilurin release or involved in other aspects such as the unusual E,Z,E triene that is formed by the same single-module PKS. However, I understand these would take a long time to answer and the authors wish to publish our current observations quickly.

Reviewer #2 (Remarks to the Author):

The manuscript has been revised based on the reviewers' comments. The authors have adequately answered the questions raised by the reviewers. Together with the additional experimental data included, the quality of the manuscript has improved and is acceptable for publication by Nature Communications.

We thank the reviewers for the comments . No further changes are required.